# Comprehensive Analysis of Chloroplast Genome of *Hibiscus sinosyriacus*: Evolutionary Studies in Related Species and Genera

**Soon-Ho Kwon** [1], **Hae-Yun Kwon** [2], **Young-Im Choi** [1] and **Hanna Shin** [1,*]

1 Department of Forest Bioresources, National Institute of Forest Science, Suwon 16631, Republic of Korea; shkwon84@korea.kr (S.-H.K.); yichoi99@korea.kr (Y.-I.C.)
2 Forest Medicinal Resources Research Center, National Institute of Forest Science, Yeongju 36040, Republic of Korea; kwonhy05@korea.kr
* Correspondence: hanna193@korea.kr; Tel.: +82-31-290-1165

**Abstract:** The *Hibiscus* genus of the Malvaceae family is widely distributed and has diverse applications. *Hibiscus sinosyriacus* is a valuable ornamental tree, but it has not been extensively researched. This study aimed to complete the chloroplast genome of *H. sinosyriacus* and elucidate its evolutionary relationship with closely related species and genera. The complete chloroplast genome of *H. sinosyriacus* was found to be 160,892 bp in length, with annotations identifying 130 genes, including 85 coding genes, 37 tRNAs, and 8 rRNAs. Interspecific variations in the *Hibiscus* spp. were explored, and *H. sinosyriacus* has species-specific single-nucleotide polymorphisms in four genes. Genome structure analysis and visualization revealed that in the *Abelmoschus* genus, parts of the large single-copy region, including *rps19*, *rpl22*, and *rps3*, have been incorporated into the inverted repeat region, leading to a duplication and an increase in the number of genes. Furthermore, within the Malvales order, the *infA* gene remains in some genera. Phylogenetic analysis using the whole genome and coding sequences established the phylogenetic position of *H. sinosyriacus*. This research has further advanced the understanding of the phylogenetic relationships of *Hibiscus* spp. and related genera, and the results of the structural and variation studies will be helpful for future research.

**Keywords:** chloroplast genome; *Hibiscus sinosyriacus*; large single-copy region; phylogenetic analysis; species differentiation; statistical analysis

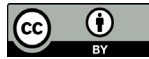

## 1. Introduction

The *Hibiscus* genus of the Malvaceae family encompasses approximately 220–250 species that are widely distributed across tropical, subtropical, and temperate climates in the form of trees, shrubs, and herbs [1]. Historically, this genus has been utilized for ornamental, culinary, and medicinal purposes [2,3]. With technological advancements and the rise of high-value industries, the applications of *Hibiscus* spp. have expanded to include indoor decoration, functional foods, cosmetics, and medicinal research [4–8]. Among the various *Hibiscus* species, *Hibiscus sinosyriacus* L. H. Bailey is a deciduous shrub native to the subtropical and tropical regions of southern China [9]. In the previous study, this species is morphologically most similar to *H. syriacus*, but with some differences, including broader leaves, elongated epicalyx tubes, and larger growth. Moreover, through an amplified fragment length polymorphism analysis, two species were clearly differentiated [10]. In the Republic of Korea, novel cultivars of *H. syriacus* have been developed by interbreeding with *H. sinosyriacus* to enhance their ornamental value, flower quality, and growth habit [11–13].

Chloroplast (cp), one of the cell organelles in plants, is integral to photosynthesis, plant immunity, and other vital biological functions, including amino acid synthesis and nitrogen metabolism [14,15]. As a symbiotic entity within cells, cp possesses ancestral

genomic DNA and is predominantly maternally inherited from angiosperms [16]. The cp genome of angiosperms is highly conserved and typically presents a quadripartite structure comprising a large single-copy (LSC), a small single-copy (SSC), and a pair of inverted repeat (IR) regions [15]. Although the cp genome remains largely intact, genetic events such as insertions, deletions, rearrangements, and copy number variations have led to plant divergence and evolution [17]. Consequently, cp genomes have been used in diverse research areas, including species differentiation, evolutionary distance estimation, parameter determination, and cultivar identification [18–20].

The evolutionary relationship between species or genera has typically been analyzed by selecting various regions within the cp genome, such as protein-coding sequences (CDSs) and mutational hotspots. Generally, well-conserved CDS regions have been primarily used for phylogenetic classification [21]. Restricted regions showing extensive variations, such as *ycf1*, *ycf3*, and *matK-trnK*, known as mutational hotspots, have been emphasized more for developing markers to identify species, cultivars, or subspecies rather than for studies determining broad evolutionary relationships between genera or between species [22–24]. Recent studies have also focused on utilizing regions outside the CDS that are well conserved but still exhibit variations for phylogenetic analysis [25].

In this study, we aimed to elucidate the complete cp genome of *H. sinosyriacus* for the first time. Additionally, by comparing in-depth the genome structure and variations with its closely related species, we hope to help in future research such as marker development. Furthermore, by clarifying the phylogenetic relationship with closely related species and genera, we aimed to confirm the evolutionary position of *H. sinosyriacus*.

## 2. Materials and Methods

### 2.1. DNA Extraction, Sequencing, Assembly, and Annotation

Fresh leaves of *H. sinosyriacus* ("Melmauve") were obtained from the Hibiscus Clonal Archive of the National Institute of Forest Science (37.15° N, 126.57° E), Suwon, Republic of Korea. Total DNA was extracted using a GeneAll® Exgene™ Genomic DNA Purification Kit (GeneAll Biotechnology, Seoul, Republic of Korea). Next-generation sequencing library construction was performed by Macrogen (Seoul, Republic of Korea) using a TruSeq™ Nano DNA Kit (Illumina, San Diego, CA, USA). Genome sequencing was performed using a NovaSeq™ 6000 platform (Illumina). The cp genome sequence was assembled using NOVOPlasity 4.3.1, an organelle assembler based on the cp genome of *H. syriacus* (KR_259989), with k-mers of 27, 29, and 35 [26]. Genes, rRNAs, tRNAs annotations, and circular maps were drawn using GeSeq (https://chlorobox.mpimp-golm.mpg.de/geseq.html, accessed on 15 March 2023) containing annotators blatN, blatX, and Chlorom [27]. Error correction was manually conducted using Sanger sequencing, by designing primers around the nucleotides where the errors occurred.

### 2.2. Comparative Analyses of cp Genome Sequences

To comprehensively compare the cp sequences of the 17 species of the Malvaceae family, we used the mVISTA program. To observe the positional changes in genes at the boundaries of each compartment structure, including hibiscus and its close relatives, we used the GeSeq annotation program to identify the boundaries of each structure.

### 2.3. Simple sequnece repeats (SSRs) Analysis

MISA version 2.1 was used to identify SSRs in the cp genomes of *H. sinosyriacus* and 16 other species, including *H. syriacus*, *H. coccineus*, *H. mutabilis*, *H. sabdariffa*, *H. rosa-sinensis*, *H. trionum*, *H. cannabinus*, *H. taiwanensis*, *Gossypium gossypioides*, *G. herbaceum*, *G. hirsutum*, *G. raimondii*, *A. esculentus*, *A. manihot*, *A. moschatus*, with *Tilia amurensis* as the outgroup [28]. The analysis was performed using parameters set at 8/mono, 3/di, 3/tri, 3/tetra, and 3/penta. Statistical analyses of average SSRs across the three genera, excluding *T. amurensis*, were conducted using R version 4.3.1 (The R Foundation, Vienna, Austria). To

evaluate the significance among group means that did not adhere to a normal distribution, we used the non-parametric Kruskal–Wallis test [29]. Subsequently, Dunn's test with Bonferroni correction was conducted for post hoc analysis [30].

### 2.4. Detection of Variants and Statistical Analyses

To compare the overall count of single-nucleotide polymorphisms (SNPs) and indels in the complete cp genomes of 16 species, using *H. sinosyriacus* as a reference, sequences were aligned using Clustal Omega version 1.2.4 [31]. Subsequently, the alignment results were subjected to pairwise comparison analysis using CLC main workbench version 23.0.2, to determine the gaps, differences, distances, percent identities, and identities of each species [32]. These metrics were calculated separately for the whole cp genome, CDSs, and specific regions, including LSC, IRa, SSC, and IRb. For the identification of SNPs and indels within the CDSs of nine species from the *Hibiscus* genus, the ClustalW alignment tool embedded in the Vector NTI Advanced 10 software was used [33]. All genes were annotated using BLAST X and Chlorom annotation engines within the GeSeq. Statistical methods were used to determine whether there were significant differences in variation across cp genomes between species and genera of the 17 species belonging to the Malvaceae family. Given that the species within each genus did not follow a normal distribution, as determined using the Shapiro–Wilk test in R version 4.3.1, we utilized the non-parametric Kruskal–Wallis test for statistical analysis. For post hoc analysis, the Dunn's test with Bonferroni correction was applied. Two methods were used to extract samples from the 17 species. The first method utilized the pairwise averages of species within each genus, whereas the second method used *H. sinosyriacus* as a reference.

### 2.5. Phylogenetic Tree Analysis

Alignment analyses of the complete cp genomes were performed using the same species as those included in the SSR analysis—nine species of *Hibiscus*, including *H. sinosyriacus*, three species of *Abelmoschus*, and four species of *Gossypium*, with *T. amurensis* as the outgroup—using Clustal Omega version 1.2.4. The following are the scientific names of the plants used for phylogenetic analysis, along with their respective GenBank accession numbers: *H. sinosyriacus* (MZ_367751), *H. syriacus* (KR_259989), *H. mutabilis* (MK_820657), *H. coccineus* (OK_336487), *H. sabdariffa* (MZ_522720), *H. rosa-sinensis* (NC_042239), *H. trionum* (OL_628829), *H. cannabinus* (NC_045873), *H. taiwanensis* (MK_937807), *A. esculentus* (NC_035234), *A. manihot* (NC_053353), *A. moschatus* (NC_053355), *G. gossypiodes* (NC_017894), *G. herbaceum* (JK_317353), *G. hirsutum* (NC_007944), *G. raimondii* (NC_016668), and *T. amurensis* (MH_169573). Phylogenetic analyses were performed separately for each region, including the whole cp genome, LSC, SSC, IR, and CDS regions. The analysis was performed using the neighbor-joining method in the CLC genomics workbench program version 23.1, with 1000 bootstraps.

## 3. Results

### 3.1. Cp Genome Assembly and Annotation of H. sinosyriacus Genes

The assembly process utilized 121,987,386 total reads, 2,315,382 of which aligned with the reference genome; 2,300,724 of these were used for assembly, with an average organelle coverage of 2173×. The complete cp genome of *H. sinosyriacus* was sequenced and assembled, resulting in a circular genome 160,892 bp in length (Figure 1). This genome was then deposited in GenBank, under the accession number MZ_367751. The genome comprised four distinct regions: LSC, IRs (IRa and IRb), and SSC. The LSC region was 89,747 bp in length, the IRa and IRb regions were 25,742 bp each, and the SSC region was 19,661 bp in length. The GC content of *H. sinosyriacus* was 36.85%. The total annotation included 130 genes, including 85 CDSs, 37 tRNAs, and 8 rRNAs (Table 1). Of the 85 CDSs, there were 12 genes in the cp genome of *H. sinosyriacus*: *petD*, *petB*, *atpF*, *ycf3*, *ndhB*, *ndhA*, *rpoC1*, *rps16*, *rps12*, *rpl16*, *rpl2*, and *clpP1*. Among these, *ycf3* had three introns and *clpP1*

had two introns, whereas the remaining 10 genes contained one intron each. Among the 37 tRNAs, eight tRNAs—*trnK-UUU, trnS-UCC, trnL-UAA, trnV-UAC*, two copies of *trnE-UUC*, and two copies of *trnA-UGC*—each contained one intron (Table 2).

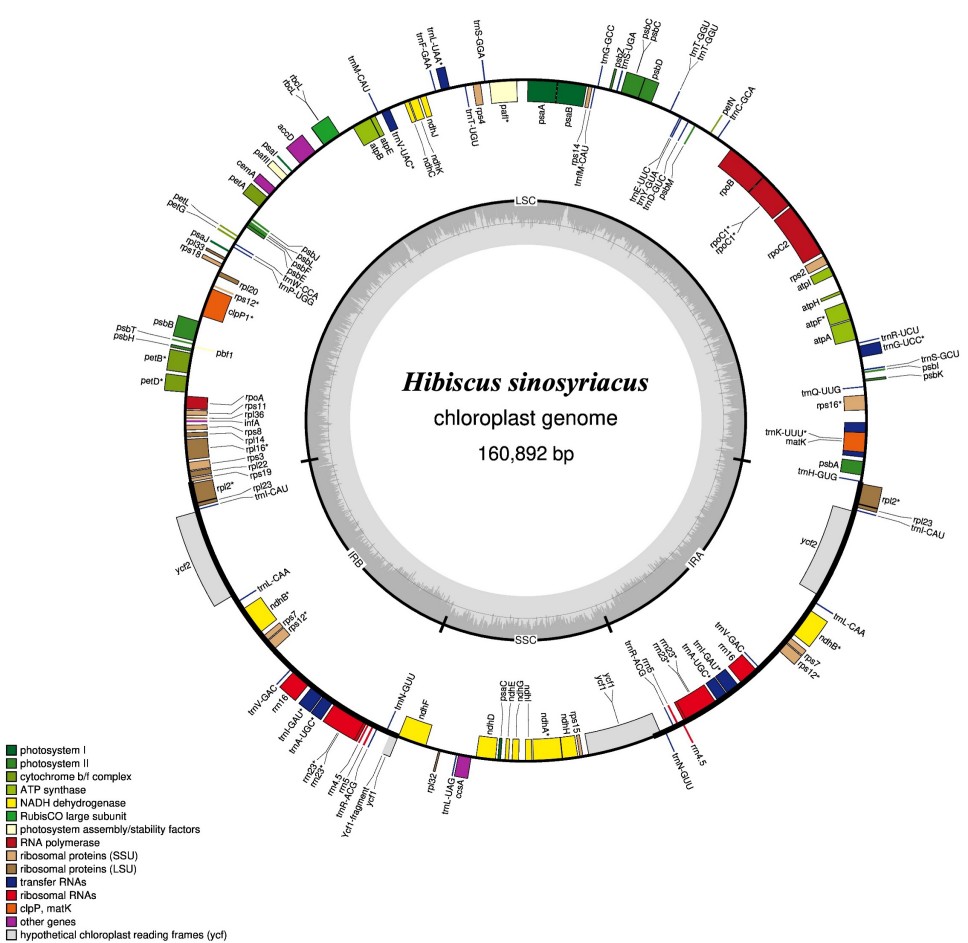

**Figure 1.** Circular map of the chloroplast genome of *H. sinosyriacus*. Genes, tRNAs, and rRNAs are presented as different-colored boxes on the outer circle. The inner circle shows the quadrant structure of the chloroplast genome. The dark gray circle shows the GC content, whereas light gray circle shows the AT content along the genome. cp, chloroplast; LSC, large single-copy; IRA, inverted repeat A; SSC, small single-copy; IRB, inverted repeat B. * indicates the presence of one and more introns.

**Table 1.** Summary of the complete cp genomes of 17 species of the Malvaceae family.

| | Genome Size (bp) | LSC | IRB | SSC | IRA | Number of Genes | Protein Coding Genes | tRNA | rRNA | GC Contents (%) |
|---|---|---|---|---|---|---|---|---|---|---|
| *H. sinosyriacus* | 160,892 | 89,747 | 25,742 | 19,661 | 25,742 | 130 | 85 | 37 | 8 | 36.85 |
| *H. syriacus* | 161,022 | 89,701 | 25,745 | 19,831 | 25,745 | 130 | 85 | 37 | 8 | 36.83 |
| *H. mutabilis* | 160,879 | 89,353 | 26,300 | 18,926 | 26,300 | 130 | 85 | 37 | 8 | 36.92 |
| *H. coccineus* | 160,280 | 89,121 | 26,243 | 18,673 | 26,243 | 130 | 85 | 37 | 8 | 36.92 |
| *H. sabdariffa* | 162,428 | 90,327 | 26,100 | 19,901 | 26,100 | 130 | 85 | 37 | 8 | 36.74 |
| *H. rosa-sinensis* | 160,951 | 89,511 | 25,597 | 20,246 | 25,597 | 130 | 85 | 37 | 8 | 36.99 |
| *H. trionum* | 160,530 | 89,272 | 26,152 | 18,954 | 26,152 | 130 | 85 | 37 | 8 | 36.90 |
| *H. cannabinus* | 162,903 | 90,351 | 26,533 | 19,486 | 26,533 | 130 | 85 | 37 | 8 | 36.65 |
| *H. taiwanensis* | 161,056 | 89,538 | 25,419 | 20,680 | 25,419 | 130 | 85 | 37 | 8 | 36.89 |
| *G. gossypiodes* | 159,959 | 88,779 | 25,588 | 20,004 | 25,588 | 129 | 84 | 37 | 8 | 37.31 |
| *G. herbaceum* | 160,140 | 88,711 | 25,604 | 20,221 | 25,604 | 129 | 84 | 37 | 8 | 37.31 |

| | | | | | | | | | |
|---|---|---|---|---|---|---|---|---|---|
| *G. hirsutum* | 160,301 | 88,817 | 25,602 | 20,280 | 25,602 | 129 | 84 | 37 | 8 | 37.24 |
| *G. raimondii* | 160,161 | 88,654 | 25,651 | 20,205 | 25,651 | 129 | 84 | 37 | 8 | 37.31 |
| *A. esculentus* | 163,121 | 88,091 | 27,999 | 19,032 | 27,999 | 133 | 88 | 37 | 8 | 36.74 |
| *A. manihot* | 163,428 | 88,214 | 28,140 | 18,934 | 28,140 | 133 | 88 | 37 | 8 | 36.70 |
| *A. moschatus* | 163,430 | 88,263 | 28,118 | 18,931 | 28,118 | 133 | 88 | 37 | 8 | 36.71 |
| *T. amurensis* | 162,564 | 91,100 | 25,493 | 20,478 | 25,493 | 129 | 84 | 37 | 8 | 36.51 |

**Table 2.** Gene contents in the cp genome of *H. sinosyriacus*.

| Role | Group of Gene | Name of Gene | No. |
|---|---|---|---|
| Photosynthesis | Photosystem I | *psaA, psaB, pasC, psaI, psaJ* | 5 |
| | Photosystem II | *psbA, psbK, psbI, psbM, psbD, psbF, psbC, psbH, psbJ, psbL, psbE, psbN, psbB* | 13 |
| | Cytochrome b/f complex | *psbT, psbZ, petN, petA, petL, petG, petD [1], petB [1]* | 8 |
| | ATP synthase | *atpI, atpH, atpA, atpF [1], atpE, atpB* | 6 |
| | Cytochrome c-type synthesis | *ccsA* | 1 |
| | Assembly/stability of photosystem I | *ycf3(pafI) [3], ycf4(pafII)* | 2 |
| | NADPH dehydrogenase | *ndhB \*[1], ndhH, ndhA [1], ndhI, ndhG, ndhJ, ndhE, ndhF, ndhC, ndhK, ndhD* | 12 |
| | Rubisco | *rbcL* | 1 |
| Transcription and translation | Small subunit of ribosome | *rpoA, rpoC2, rpoC1 [1], rpoB, rps16 [1], rps2, rps14, rps4, rps18, rps12 \*\*\*[1], rps11, rps8, rps3, rps19, rps7 \*, rps15* | 18 |
| | Large subunit of ribosome | *rpl33, rpl20, rpl36, rpl14, rpl16 [1], rpl22, rpl2 \*[1], rpl23 \*, rpl32* | 11 |
| | Translational initiation factor | *infA* | 1 |
| | Ribosomal RNA | *rrn16 \*, rrn4.5 \*, rrn5 \*, rrn23 \** | 8 |
| | Transfer RNA | *trnH-GUG, trnK-UUU [1], trnQ-UUG, trnS-GCU, trnS-UCC [1], trnR-UCU, trnC-GCA, trnD-GUC, trnY-GUA, trnE-UUC [1]\*\*, trnI-GGU, trnS-UGA, trnG-UCC, trnfM-CAU \*\*, trnS-GGA, trnT-UGU, trnL-UAA [1], trnF-GAA, trnV-UAC [1], trnW-CCA, trnP-GGU, trnL-CAA \*, trnV-GAC \*, trnA-UGC [1]\*, trnR-ACG \*, trnN-GUU \*, trnL-UAG, trnI-CAU* | 37 |
| Other | RNA processing | *matK* | 1 |
| | Carbon metabolism | *cemA* | 1 |
| | Fatty acid synthesis | *accD* | 1 |
| | Proteolysis | *clpP1 [2]* | 1 |
| | Component of TIC complex | *ycf1* | 1 |
| | Hypothetical proteins | *ycf2 \** | 2 |
| Total number of genes | | | 130 |

[1] Contained one intron in the gene. [2] Contained two introns in the gene. [3] Contained three introns in the gene. * There are two copies in the genome. ** There are three copies in the genome. *** Trans-spliced gene.

### 3.2. Comparative Structural Analysis

The positions of the genes at the boundaries of each quadripartite structure of the cp genome play a crucial role in observing insertions, deletions, and structural transformations in large frames [34,35]. The structural differences in 17 species, including 9 species

of the genus *Hibiscus*, were analyzed to determine the gene loci at the beginning and end of the structural boundaries (Figure 2). The average sizes of the cp genomes were 161,216 bp for the *Hibiscus* genus, 163,326 bp for the *Abelmoschus* genus, 160,140 bp for the *Gossypium* genus, and 162,564 bp for *T. amurensis*. Among them, the *Abelmoschus* genus had the largest genome, whereas *Gossypium* had the smallest. The average sizes of the LSC region in the genera *Hibiscus* and *Gossypium* were 89,268 and 89,658 bp, respectively, whereas that in the genus *Abelmoschus* was 88,189 bp, with a difference of 1079–1469 bp. Conversely, the genus *Abelmoschus* exhibited an expansion in the IR region by 1849–2593 bp, compared with the other two genera, which was due to the presence of *rps19*, *rpl22*, and *rps3* genes. In the LSC region, the *rps19* gene (excluding the genus *Abelmoschus*) spanned the boundary of seven species: *H. sinosyriacus*, *H. syriacus*, *H. sabdariffa*, *H. cannabinus*, *G. gosyspioides*, *G. herbaceum*, and *G. raimondii*. Additionally, the *rps16* gene in the three species of the *Abelmoschus* genus also crossed the boundary line. In contrast, there were seven species for which no genes were located across the boundary: *H. mutabilis*, *H. coccineus*, *H. rosa-sinensis*, *H. trionum*, *H. taiwanensis*, *G. hirsutum*, and *T. amurensis*. Notably, *H. rosa-sinensis* exhibited the largest distance between the gene and the boundary (103 bp). In the IRb structure, three species from the *Abelmoschus* genus began with the *rps3* gene, whereas in the other species, the *rpl2* gene appeared after the boundary with the LSC region. Within the *Hibiscus* genus, the end of the *ycf1* fragment in *H. sabdariffa* was 489 bp before the start of the small SSC region, a pattern similar to that observed in the *Abelmoschus* genus. The *ycf1* fragment of *H. coccineus* showed a 116 bp difference from the SSC boundary. All other *ycf1* fragments of *Hibiscus* species crossed the SSC boundary. In most cases, the last gene of the IRa, *rpl2*, was located 57–137 bp before the LSC boundary, whereas in the case of the *Abelmoschus* genus, the *rpl16* gene was present.

The GC content of *H. sinosyriacus* was 36.85%, and the average GC content of the genus *Hibiscus* was ~36.85%. The average GC content of the genus *Abelmoschus* was 36.72%, which was lower than that of the genus *Hibiscus*, whereas that of the genus *Gossypium* was 37.29%, which was much higher. The GC content of *T. amurensis* was the lowest (36.51%) (Table 1). Differences were observed in the number of genes in the cp genome among the genera. The genus *Abelmoschus* had three additional genes compared with other genera, as the IR region included *rps19*, *rpl22*, and *rps3*. Unlike other genera, the genus *Abelmoschus* experienced an increase in gene count owing to the incorporation of *rps19*, *rpl22*, and *rps3* from the existing LSC region into the IR region, resulting in duplication. This led to a three-fold increase in the gene count. Although both synonymous and non-synonymous SNPs were observed within the *infA* gene of plants of the *Hibiscus* and *Abelmoschus* genera, they were well preserved within the cp genome. However, although a deletion of 7 bp within the *infA* gene in the *Gossypium* genus led to the appearance of a premature stop codon, an insertion of 11 bp resulted in a premature stop codon, confirming the loss of the *infA* gene in *T. amurensis* (Figure 3).

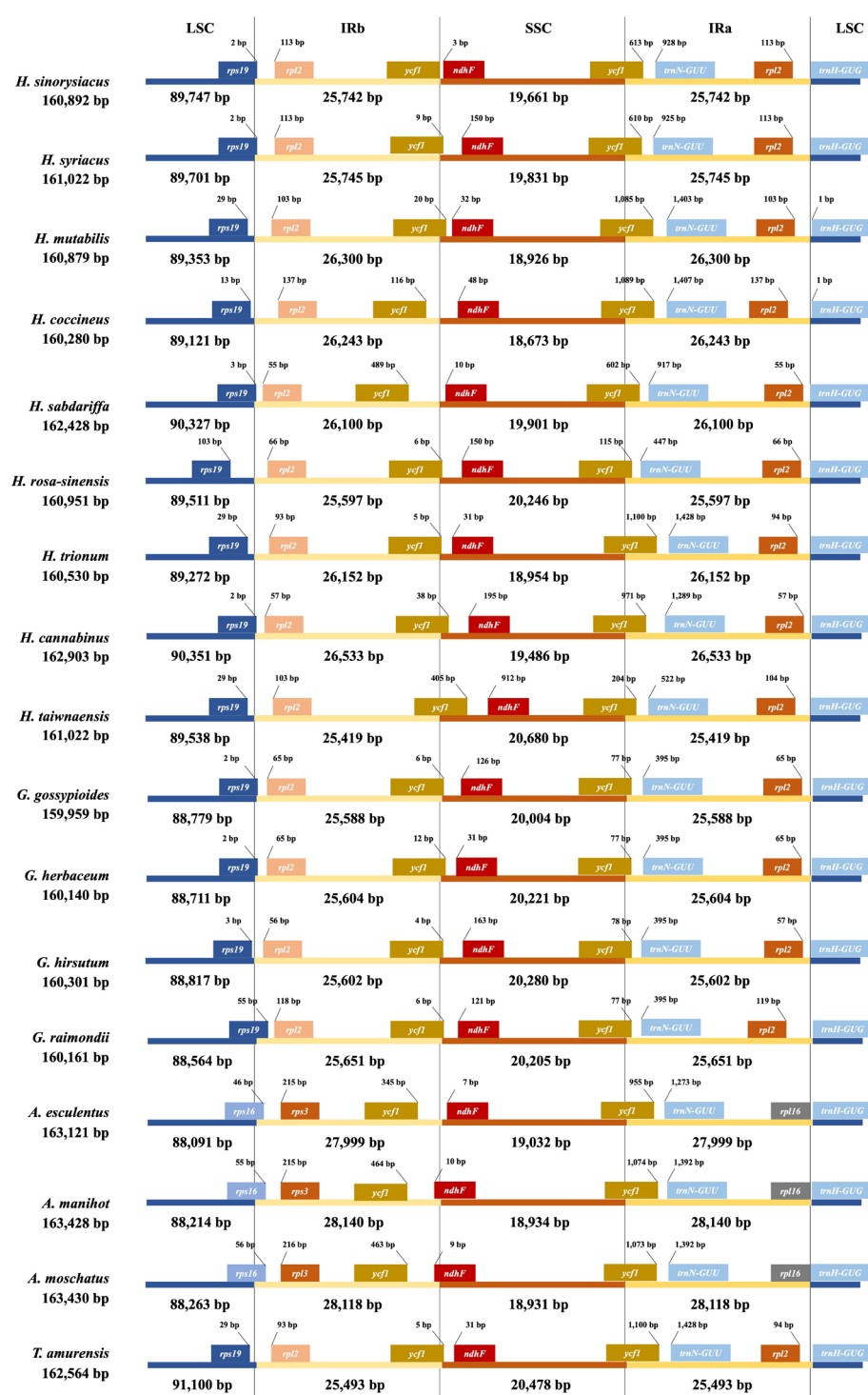

**Figure 2.** Distance between adjacent genes and junctions of the SSC, LSC, and two IR regions among plastid genomes of 17 species.

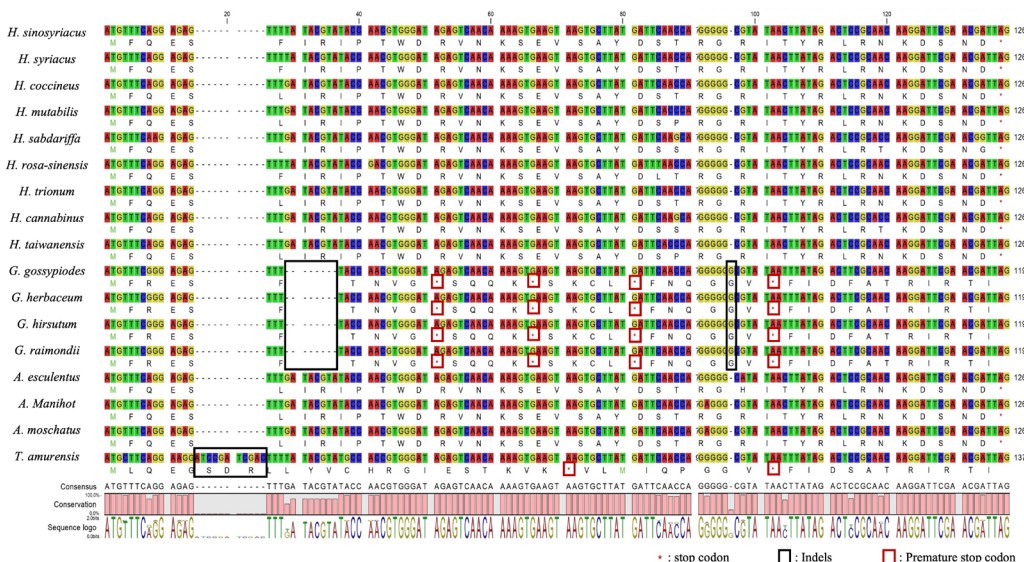

**Figure 3.** Structural variations in the *infA* gene among different genera in the Malvaceae family.

### 3.3. SSRs Analysis

The total number of SSRs in *H. sinosyriacus* was 686, comprising 192 mononucleotides, 408 dinucleotides, 76 trinucleotides, 6 tetranucleotides, and 4 pentanucleotides. A analysis of SSRs across 17 species, including *H. sinosyriacus*, revealed an average total SSR count of 692.35. The species with the highest number of SSRs was *A. esculentus* (748), whereas the one with the lowest was *G. raimondii* (633). Among these, *Abelmoschus* had the highest average SSR count (745.33), followed by *T. amurensis* (735), *Hibiscus* (689), and *Gossypium* (649.5). Notably, within the *Hibiscus* genus, *H. rosa-sinensis* had a particularly low SSR count (653). The distributions of total SSRs and monopenta-SSR motifs were not proportional across species. *A. moschatus* had the highest number of mononucleotide SSRs, whereas *H. rosa-sinensis* had the lowest. *A. esculentus* had the most dinucleotide SSRs, whereas *G. raimondii* had the least. *A. manihot* had the highest number of trinucleotide SSRs, whereas *G. hirsutum* had the lowest. With respect to the tetranucleotide SSRs, *A. manihot* had the highest count, whereas *H. sinsyriacus*, *H. rosa-sinensis*, and *G. hirsutum* had only six. *T. amurensis* had the highest count of pentanucleotides (11), whereas that in the other species ranged between 1 and 4. The distribution of SSRs was also analyzed according to region. SSRs in the LSC region accounted for ~63.3% of the total, whereas those in the SSC and IR regions accounted for 11.7% and ~25% of the total, respectively. The LSC region showed an SSR distribution pattern that was most similar to that of the overall genome of the 17 species, whereas the SSC region displayed a distinct pattern. In particular, *H. sinosyriacus* and *H. syriacus* had a notably higher number of SSRs than the other species, whereas pentanucleotide repeats were absent in all species. The IR regions had a distribution pattern that was more similar to the SSR distribution of the overall genome than that of the SSC region, but with a higher proportion of dinucleotides. Tetranucleotide repeats were absent in all the species, except *T. amurensis* (Figure 4). In the comparison of differences in the number of SSRs among the three closely related genera excluding *T. amurensis*, the quantity of SSRs in the *Hibiscus* genus was intermediate between the other two genera and showed no significant difference from them. However, there was a significant difference between *Abelmoschus*, which had the most SSRs, and *Gossypium*, which had fewer SSRs (Table 3).

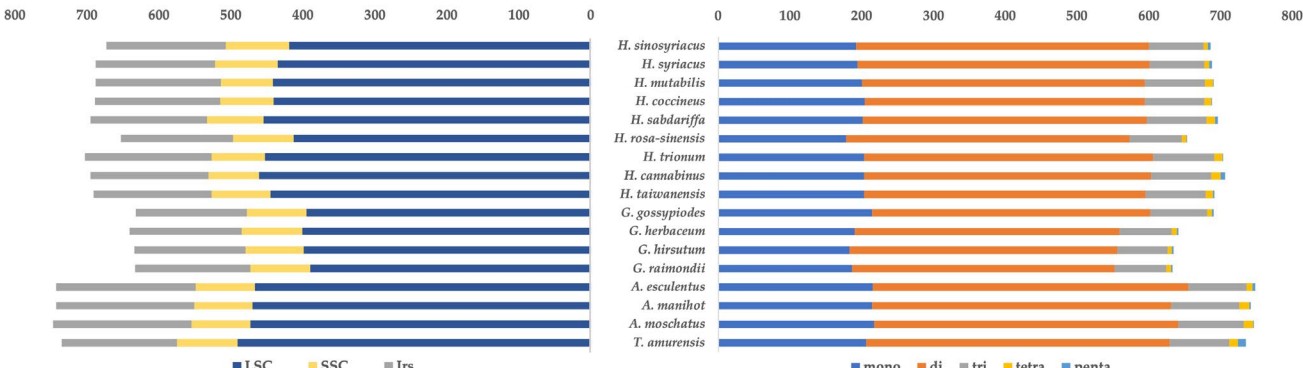

**Figure 4.** SSR analysis in 17 species of the Malvaceae family. (**Left**), region-specific SSRs; (**Right**), SSRs presented by repeat motif lengths.

**Table 3.** Statistical analysis of variants and differences in the cp genomes of *Hibiscus sinosyriacus* and related species.

| Variants Type | Kruskal–Wallis Test | Post hoc Analysis | | |
|---|---|---|---|---|
| | *p*-Value | H-G | G-A | A-H |
| SSRs | $7.39 \times 10^{-3}$ ** | $1.32 \times 10^{-1}$ | $2.60 \times 10^{-3}$ ** | $6.04 \times 10^{-2}$ |
| Identities | $4.64 \times 10^{-8}$ *** | $9.95 \times 10^{-7}$ *** | 1.00 | $6.64 \times 10^{-4}$ *** |
| Differences | $8.52 \times 10^{-3}$ ** | $3.56 \times 10^{-2}$ * | $4.42 \times 10^{-3}$ ** | $3.24 \times 10^{-1}$ |
| Gaps | $1.39 \times 10^{-2}$ * | $2.30 \times 10^{-2}$ * | $1.17 \times 10^{-2}$ * | $6.87 \times 10^{-1}$ |
| Gaps and differences | $1.21 \times 10^{-2}$ * | $2.67 \times 10^{-2}$ * | $8.55 \times 10^{-3}$ ** | $5.46 \times 10^{-1}$ |

The non-parametric Kruskal–Wallis test was used to assess significance, followed by post hoc analyses using the Dunn's test with Bonferroni correction for multiple comparisons. H, *Hibiscus* genus; G, *Gossypium* genus; A, *Abelmoschus* genus. *** $p < 0.001$, ** $0.001 \le p < 0.01$, * $0.01 \le p < 0.05$.

*3.4. Comparative Sequence Identification Analysis via Visualization*

The mVISTA program, which visualizes the similarity of comparative sequences, was used to understand these differences intuitively [36]. We explored the sequence variations in 17 species using *H. sinosyriacus* as a reference (Figure 5). Generally, sequence differences are observed more frequently in non-coding regions than in coding regions. In the non-coding regions, significant differences were observed within the intron regions of *matK-atpA*, *atpF-atpI*, *rpoB-psbD*, *psbC-psaB*, *rps4-ndhJ*, *ndhC-atpE*, *atpB-rbcL*, *pafII-cemA*, *petA-psbJ*, and *clpP1-rpl16*. These differences were predominantly distributed in the LSC region. In the coding regions, differences were frequently found within genes such as *rpoC2*, *rpoB*, *pafI*, *ycf2*, *ycf1*, and *ndhF*. In the SSC region, differences were observed between the *ndhF* and *ccsA* genes and within the intron region of *ndhA*. In the IR region, differences were observed between *rps12* and *trnV-GAC*. The location of the *ycf1* gene exhibited two distinct patterns across species. The first pattern showed a portion of the *ycf1* gene initiated at the start of the SSC region in the forward direction, with the entire sequence of the *ycf1* gene located in the reverse direction at the end of the SSC. This pattern was observed in eight species of the *Hibiscus* genus, excluding *H. rosa-sinensis*, and in three species of the *Abelmoschus* genus. Conversely, the second pattern, distinct from the first, lacked the partial *ycf1* gene at the beginning of the SSC but contained the complete sequence in the reverse direction at the end. The latter pattern was characteristic of four species from the *Gossypium* genus and *T. amurensis*. Notably, *H. rosa-sinensis* deviated from the first pattern, where the species typically had an *ycf1* partial sequence spanning 500–600 bp. Instead, *H. rosa-sinensis* replicated only a short 113 bp partial sequence at the beginning. Additionally, upon examining the sequence identity patterns, it was evident that the patterns were

either grouped by genus or varied distinctly. Species-specific patterns were also observed. For instance, a unique pattern was identified between the *atpF* and *atpH* introns in *H. syriacus* and within the IR region between *rps12* and *trnV-GAC* intron in *H. trionum*. Excluding *H. syriacus*, the remaining species showed similar patterns of variation; however, unique patterns were often observed, depending on the species. Notably, *H. syriacus* showed much less difference from *H. sinosyriacus* than the other species. Unlike other species, these two species can be crossbred and have flower shapes similar to those of shrubs. Noticeable differences between the two species were observed in the non-coding regions of *atpF* and *atpH*, *psbZ* and *rps14*, *accD* and *psaI*, *petA* and *psbJ*, *rps18* and *rpl20*, *rps12* and *trnV-GAC*, *rpl32* and *trnL-UAG*, and so on. *H. mutabilis* had a different pattern of sequence similarity between *atpF* and *atpH* and *trnR-ACG* and *trnN-GUU*, as compared with the other species. *H. rosa-sinensis* showed large differences in the *rpoC2*, *ycf2*, and *ycf1* partial genes. In terms of sequence similarity, *H. sinosyriacus*, *H. syriacus*, *H. mutabilis*, *H. coccineus*, *H. sabdariffa*, and *H. cannabinus* exhibited similar patterns.

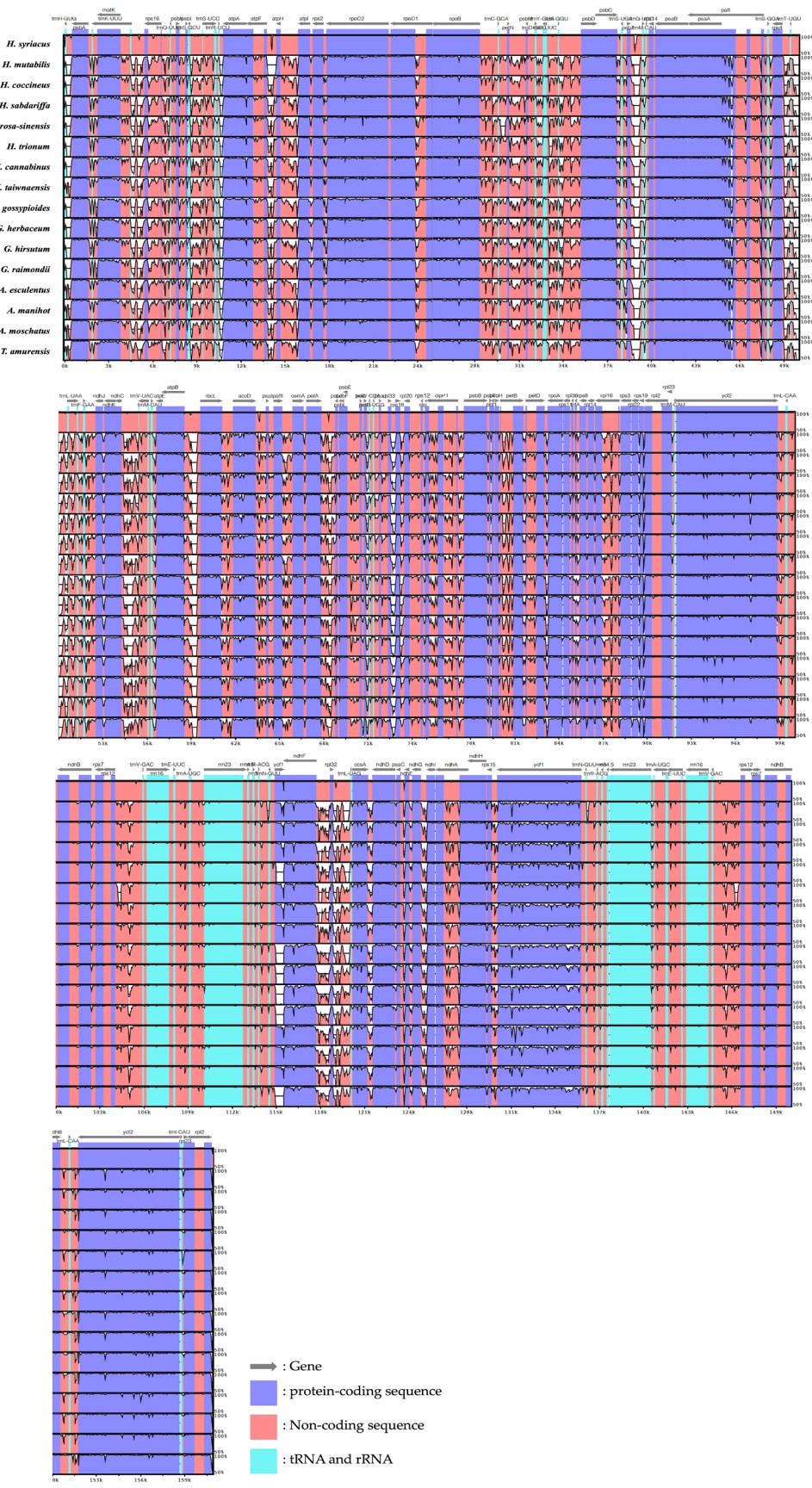

**Figure 5.** Visualization of alignment identity among 16 species. Sequences were annotated and identified using different colors. Sequence identity ratio has been presented through vertical depth, using *H. sinosyriacus* as a reference.

*3.5. Comparison Analysis of Pairwise Heatmap*

From the pairwise analysis results of 17 species in the Malvaceae family, we confirmed that they are well grouped by genus. First, the heatmaps of whole genome, within the *Hibiscus* genus, the combinations of *H. sinosyriacus* and *H. syriacus* (99.72% and 2.43 × $10^{-4}$), and *H. taiwanensis* and *H. mutabilis* (99.81% and 3.23 × $10^{-4}$) showed similar values for similarity and distance, respectively. *H. coccineus* showed close values with *H. mutabilis* (97.59% and 5.01 × $10^{-3}$), *H. trionum* (96.99% and 5.47 × $10^{-3}$), and *H. taiwanensis* (97.56% and 5.20 × $10^{-3}$). In the *Gossypium* genus, *G. gossypioides* and *G. reimondii* had the closest distance of 2.79 × $10^{-3}$, whereas *G. herbaceum* and *G. reimondii* were the most similar (99.23%). In the *Abelmoschus* genus, *A. manihot* and *A. moschatus* were the closest, at 99.95% similarity. The distances between *A. esculentus* and *A. manihot* and between *A. esculentus* and *A. moschatus* were both 6.95 × $10^{-4}$. The pairwise heatmap for "Gaps and differences" on the right appeared to be proportional to the sequence similarity on the left. Following this, upon examining the heatmaps for CDS, the identity and distance in relation to CDS displayed a pattern similar to that of the whole genome. The interspecies sequence similarity did not show significant differences when compared with those of the whole genome. As CDS sequences are better conserved than non-coding sequences, the sequence similarity for most species increased. However, the similarity between the combination of *H. sinosyriacus* and *H. syriacus* was observed to decrease to 99.59%, compared with 99.72% in the whole genome (Figure 6). Upon analyzing the sequence similarity differences among the three genera, contrary to the results from the SSR analysis, there was no significant difference between the *Gossypium* and *Abelmoschus* genera. However, there was a substantial difference between these two genera and *Hibiscus*, with values of 9.95 × $10^{-7}$ and 6.64 × $10^{-4}$, respectively. Furthermore, in the indices for gaps and differences, there was no significant difference between *Abelmoschus* and *Hibiscus*. However, significant differences were observed in the other two combinations, *Hibiscus* and *Gossypium* (2.67 × $10^{-2}$) and *Gossypium* and *Abelmoschus* (8.55 × $10^{-3}$) (Table 3).

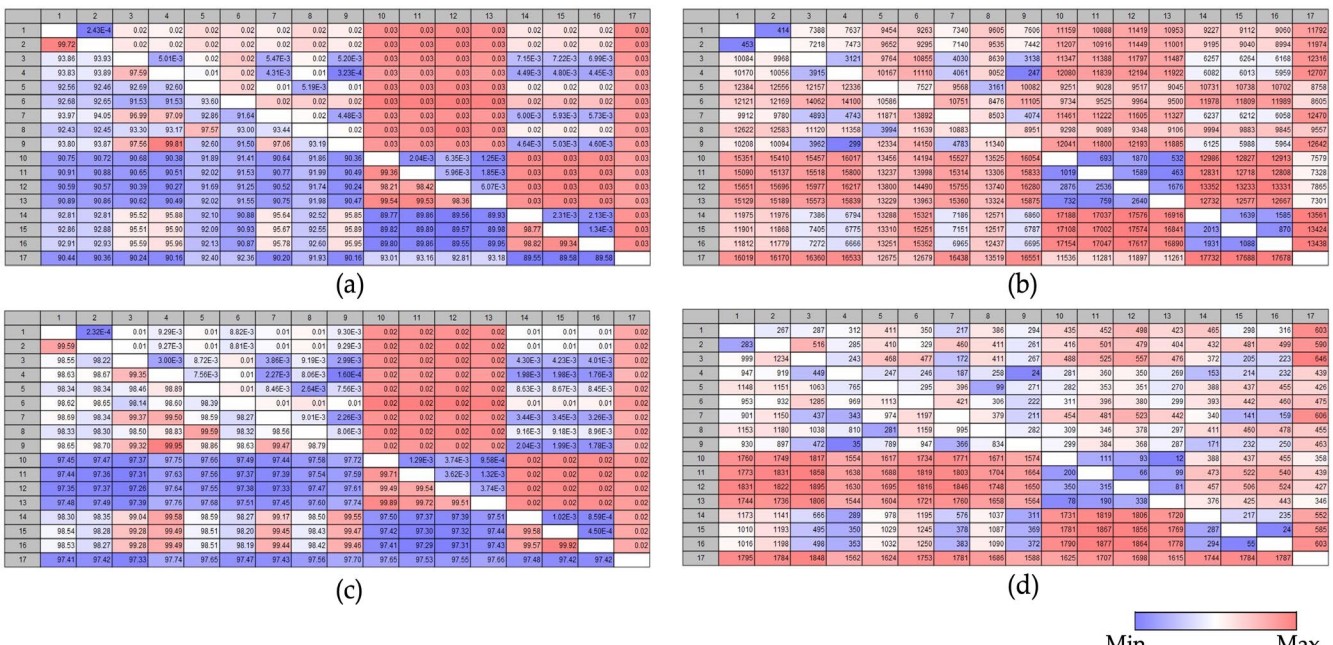

**Figure 6.** Pairwise comparison heatmap. (**a**) Percent identities and distances of whole genomes, (**b**) gaps and differences in whole genomes, (**c**) percent identities and distances of CDSs, and (**d**) gaps

and differences in CDSs. (**a,c**) Top, distances; bottom, percent identities. (**b,d**) Top, differences; bottom, gaps. CDS, coding sequence. 1, *H. sinosyriacus*; 2, *H. syriacus*; 3, *H. coccineus*; 4, *H. mutabilis*; 5, *H. sabdariffa*; 6, *H. rosa-sinensis*; 7, *H. trionum*; 8, *H. cannabinus*; 9, *H. taiwanensis*; 10, *G. gossypioides*; 11, *G. herbaceum*; 12, *G. hirsutum*; 13, *G. raimondii*; 14, *A. esculentus*; 15, *A. manihot*; 16, *A. moschatus*; 17, *T. amurensis*.

### 3.6. Exploration of Variants in the CDS of Hibiscus spp.

In this comprehensive study of the cp genome of *H. sinosyriacus*, we successfully assembled it for the first time and explored its evolutionary relationship with eight closely related species, by examining variations within the CDS. Using *H. sinosyriacus* as the reference, 130 genes were examined. Notably, 36 genes showed no variations. These genes included 28 tRNA genes (such as *trnH-GUG*, *trnK-UUU*, *trnQ-UUU*, *trnS-GCU*, *trnG-UCC*, *trnC-GCA*, *trnY-GUA*, *trnG-UCC*, *trnM-CAU*, *trnS-GGA*, *trnF-GAA*, *trnM-CAU*, *trnW-CCA*, *trnM-CAU*, *trnL-CAA*, *trnV-GAC*, *trnI-GAU*, *trnA-UGC*, *trnR-ACG*, *trnN-GUU*, *trnL-UAG*, *trnN-GUU*, *trnR-ACG*, *trnA-UGC*, *trnL-GAU*, *trnV-GAC*, *trnL-CAA*, and *trnI-CAU*), 2 rRNA genes (*rrn5*), and 6 other genes. In an analysis of variants across different species, several distinct patterns emerged. In *H. sinosyriacus*, species-specific SNPs were identified in the CDS regions of the *matK*, *psbC*, *ndhK*, and *ycf2* genes, with one SNP detected for each gene, totaling four SNPs. *H. syriacus* had only 4 species-specific SNPs, 6 common SNPs, and 3 species-specific inserts, resulting in a total variant count of 13. *H. coccineus* displayed 136 species-specific SNPs, 608 common SNPs, and a combined total of 198 indels, resulting in 942 variants. Both *H. mutabilis* and *H. taiwanensis* showed 4 species-specific SNPs, 655 common SNPs, and 815 and 821 total variants, respectively. *H. sabdariffa* had 76 species-specific SNPs, 708 common SNPs, and 1083 variants. *H. rosa-sinensis* contained 326 species-specific SNPs, 311 common SNPs, and 877 variants. *H. trionum* had 89 species-specific SNPs, 624 common SNPs, and 927 variants. Finally, *H. cannabinus* had 101 species-specific SNPs, 705 common SNPs, and 1067 variants. Indel regions were identified in the following 13 genes: *matK*, *rpoB*, *atpB*, *rbcL*, *rpl20*, *rpl23*, *ccsA*, *rpoC2*, *rps14*, *accD*, *ycf2*, *ndh5*, and *ycf1*. Species-specific indels were observed in several genes. The gene *rpoC2* exhibited a species-specific insert exclusive to *H. rosa-sinensis*. Similarly, *rps14* exhibited a species-specific insertion in *H. rosa-sinensis*. The *accD* gene revealed species-specific indels in *H. trionum* and a unique insert in *H. rosa-sinensis*. The *ycf2* gene displayed general indels with species-specific inserts in *H. trionum* and *H. rosa-sinensis*. The *ndh5* gene had species-specific indels in *H. sabdariffa*, *H. trionum*, and *H. rosa-sinensis*. Finally, the *ycf1* gene presented general indels and species-specific indels in *H. syriacus*, *H. rosa-sinensis*, and *H. coccineus*. The *ycf1* gene is particularly notable for its extensive variation. It harbored a diverse range of indels, especially between positions 5688 and 5742 bp, and was densely populated with species-specific SNPs and indels. Intriguingly, although *H. sinosyriacus* and *H. syriacus* exhibited significant similarities, a unique indel specific to *H. syriacus* was identified in this region. Among the 13 genes analyzed, *matK*, *rpoB*, *atpB*, *rbcL*, *rpl20*, *rpoC2*, and *rps14* were located in the LSC region, whereas *accD*, *ccsA*, *ndh5*, and *ycf1* were located in the SSC region, and *rpl23* and *ycf2* were duplicated and present in the IR regions (Table 4).

In the analysis of stop codon usage across various species, we examined the termination codons in 85 genes (Table S1). The distribution of stop codons was as follows: TAA, 55.95%; TGA, 20.78%; TAG, 23.27%. Among the nine *Hibiscus* spp. analyzed, variations in stop codons were observed for five genes: *atpB*, *accD*, *petA*, *rpl16*, and *ccsA* (Table 5). Specifically, for the *atpB* gene, *H. sabdariffa* and *H. cannabinus* both utilized TAG, whereas the remaining seven species used TGA. In the case of the *accD* gene, only *H. trionum* had TAA, whereas the other eight species used TAG. In the case of the *petA* gene, both *H. sinosyriacus* and *H. syriacus* used TAA, whereas the other seven species used TAG. In the case of the *rpl16* gene, *H. coccineus* was the only species with TAA, with TAG being prevalent in the other eight species. Finally, in the case of the *ccsA* gene, only *H. rosa-sinensis* had TAA, whereas TGA was observed in the remaining eight species. The distribution of the

variations was as follows: TAG to TGA in one instance, TAA to TAG in three instances, and TAA to TGA in one instance.

**Table 4.** Summary of variation among CDS of nine *Hibiscus* spp.

| Name | SNP | | | Indels | | | | | Variants Total |
|---|---|---|---|---|---|---|---|---|---|
| | Species-Specific SNP | Common | Total SNP | Species-Specific Insert | Species-Specific Deletion | Common Insert | Common Deletion | Total Indel | |
| *H. sinosyriacus* | 4 | – | 4 | – | – | – | – | – | 4 |
| *H. syriacus* | 4 | 6 | 10 | 3 | – | – | – | 3 | 13 |
| *H. coccineus* | 136 | 608 | 744 | 6 | 18 | 99 | 75 | 198 | 942 |
| *H. mutabilis* | 4 | 655 | 659 | – | – | 87 | 69 | 156 | 815 |
| *H. sabdariffa* | 76 | 708 | 784 | 19 | 15 | 156 | 109 | 299 | 1083 |
| *H. rosa-sinensis* | 326 | 311 | 637 | 102 | 15 | 51 | 72 | 240 | 877 |
| *H. trionum* | 89 | 624 | 713 | 26 | 8 | 105 | 75 | 214 | 927 |
| *H. cannabinus* | 101 | 705 | 806 | 6 | 2 | 144 | 109 | 261 | 1067 |
| *H. taiwanensis* | 4 | 655 | 659 | – | – | 87 | 75 | 162 | 821 |

**Table 5.** Variation of stop codons among genes of nine *Hibiscus* spp.

| Gene Name | *H. sinosyriacus* | *H. syriacus* | *H. coccineus* | *H. mutabilis* | *H. sabdariffa* | *H. rosa-sinensis* | *H. trionum* | *H. cannabinus* | *H. taiwanensis* |
|---|---|---|---|---|---|---|---|---|---|
| *atpB* | TGA | TGA | TGA | TGA | TAG | TGA | TGA | TAG | TGA |
| *accD* | TAG | TAG | TAG | TAG | TAG | TAG | TAA | TAG | TAG |
| *petA* | TAA | TAA | TAG | TAG | TAG | TAG | TAG | TAG | TAG |
| *rpl16* | TAG | TAG | TAA | TAG | TAG | TAG | TAG | TAG | TAG |
| *ccsA* | TGA | TGA | TGA | TGA | TGA | TAA | TGA | TGA | TGA |

*3.7. Comparative Phylogenetic Analyses*

We performed a comparative analysis of phylogenetic trees derived from both the whole cp genome and CDS regions (Figure 7). The results from the pairwise heatmap analysis displayed minor differences between the whole genome and CDS. However, the comparative outcomes utilizing both phylogenetic trees were almost identical. Using 17 species from four genera of the Malvaceae family, we investigated the evolutionary process of *H. sinosyriacus*. Among the four genera, *T. amurensis*, which was anticipated to have the greatest genetic distance, appropriately diverged early as an outgroup. Subsequently, the *Gossypium* genus differentiated earlier than the other two genera. *H. sinosyriacus*, *H. syriacus*, and *H. rosa-sinensis* diverged earlier from other species, with *H. sinosyriacus* and *H. syriacus* displaying a monophyletic relationship. This was followed by the divergence of *H. cannabinus* and *H. sabdariffa*, both showing a monophyletic structure. *H. coccineus* and *H. trionum* sequentially differentiated in a paraphyletic manner. The remaining species of the *Hibiscus* genus, *H. mutabilis* and *H. taiwanensis*, diverged with the species of the *Abelmoschus* genus in a monophyletic pattern.

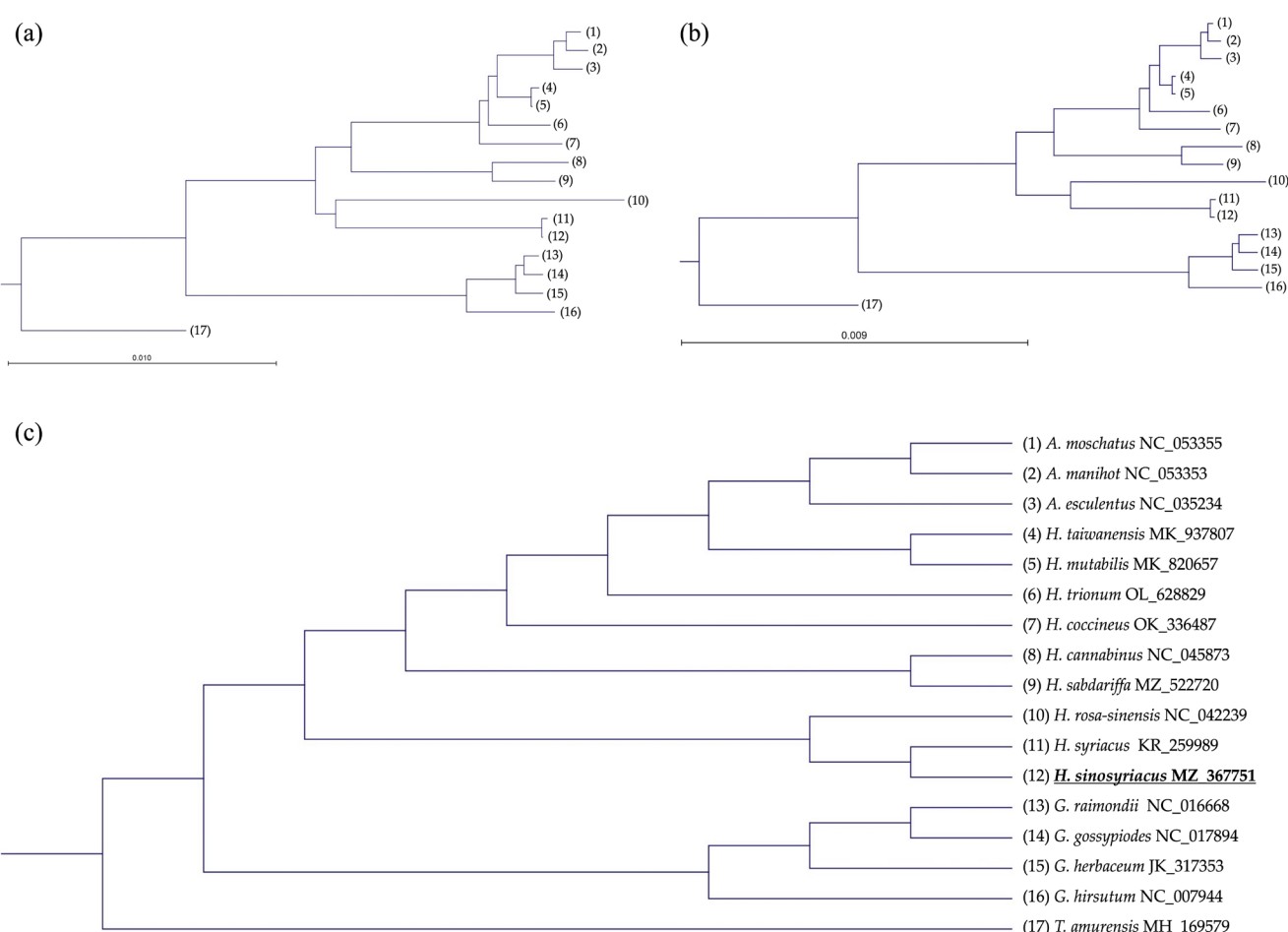

**Figure 7.** Phylogenetic analysis of 17 species of the Malvaceae family. (**a**) Phylogenetic tree derived from whole cp genome, (**b**) phylogenetic tree derived from CDS, and (**c**) cladogram. Phylograms were drawn using the maximum likelihood method, with the CLC main workbench version 23.0.2 program. Bootstrap values derived from 1000 pseudo replicates were indicated near the nodes. The numbers at the tips of the branches in phylogenetic trees (**a**,**b**) correspond to the numbered species in the cladogram (**c**).

## 4. Discussion

This study successfully executed the assembly of the cp genome of *H. sinosyriacus*, and the data obtained from this study offer profound insights into the structure and content of the cp genome of this species. Notably, the lengths and compositions of the four distinct regions of the genome—LSC, IRs (IRa and IRb), and SSC—were consistent with those of many other angiosperm cp genomes. Additionally, the number of intrinsic genes and tRNAs found in the *H. sinosyriacus* cp genome underscores its complexity and diversity. The number of genes containing introns and their locations can indicate the structural characteristics and evolutionary significance of the genome.

An analysis of the cp genome structure of *H. sinosyriacus* and other related species provides crucial information for elucidating the structural differences and features of each genome. Gene positions at the boundaries between the four distinct regions of each genome play a pivotal role in observing large-frame insertions, deletions, and structural alterations within the genome. Specifically, the genome of the genus *Abelmoschus* showed an increase in the number of genes owing to the expansion of the IR region and the inclusion of *rps19*, *rpl22*, and *rps3*. Such changes may be associated with the movement or replication of specific genes during genome evolution. Moreover, the *infA* gene is the most

mobile cp gene known in plants, and species of the genera *Gossypium* and *Tilia* have probably repeatedly transferred the *infA* gene from the cp to the nucleus for functional or evolutionary reasons [37]. Additionally, as plants evolve into higher plants, the *infA* gene tends to disappear from the cp. Although the gene is reported to be almost absent in the Malvales order, it has been confirmed that it remains intact in the cp of many higher plants, including those in the *Hibiscus* and *Abelmoschus* genera [17].

An analysis of SSRs in the cp genomes of 17 species, including *H. sinosyriacus*, provides vital information for understanding the structural characteristics and evolutionary patterns of the genome. The total number and distribution of SSRs varied between species and genera. In particular, *Abelmoschus* had the highest average number of SSRs. Such differences may stem from the evolutionary background and structural changes in the genomes of each genus. Statistical analysis revealed significant differences in SSR distribution among the three genera, interpreted as reflecting the evolutionary characteristics and genomic structural variability of each genus. In particular, the differences between *Abelmoschus* and *Gossypium* may be related to the evolutionary distance between the two genera. These findings offer valuable insights into plant evolution and diversity through SSR analyses of the cp genome.

Visualization analysis using the mVISTA program clearly delineated the cp genome sequence differences among 17 species, including *H. sinosyriacus*. Generally, sequence differences are observed more frequently in non-coding regions than in coding regions. In particular, the differences in the LSC region were noteworthy. In the coding regions, differences were frequently observed within specific genes, potentially reflecting evolutionary differences between species. Additionally, the position of the *ycf1* gene exhibited two distinct patterns depending on the species. These differences in pattern may be related to the evolutionary background [22]. The similarities between *H. sinosyriacus* and *H. syriacus* align with the fact that the two species can be interbred and have similar flower morphologies. Furthermore, the unique sequence similarity patterns observed only in specific species may indicate a unique evolutionary background of this species.

Through a pairwise heatmap analysis of the overall identity and distance of the cp genome, clear genetic differences among the three genera, *Hibiscus*, *Gossypium*, and *Abelmoschus*, were identified. Color-coded clustering facilitates an intuitive understanding of sequence similarities and differences between species. Specifically, the *Hibiscus* genus showed significant genetic differences in most regions, compared with the other two genera. However, no significant differences were observed between *Abelmoschus* and *Gossypium*. These findings suggest that the *Hibiscus* genus may have unique evolutionary characteristics compared to the other two genera.

By studying the cp genome of *H. sinosyriacus*, evolutionary relationships with eight species were successfully explored, focusing on variations within the CDS. In this study, 130 genes were reviewed, with no variations found in 36 genes. These results suggested that certain genes maintained stable characteristics throughout the evolutionary process. Several unique patterns emerged in the analysis of variation among various species. Notably, a high similarity was observed between *H. syriacus* and *H. sinosyriacus*, but species-specific indels were found in *H. syriacus*. These results indicated that despite the close relationship between the two species, each has unique evolutionary characteristics. Additionally, species-specific indels were observed in each species, especially in the *ycf1* gene, where various indels, as well as species-specific SNPs and indels, were densely distributed. These results indicated that the *ycf1* gene underwent various mutations during the evolutionary process. An extensive analysis of stop codon usage confirmed the distributions of TAA, TGA, and TAG. For specific genes, there were species-specific differences in stop codon usage, which might be related to the genetic characteristics [38].

In this study, the comparative analysis of the phylogenetic tree based on the whole cp genome and the CDS region provides a crucial key to deeply understanding the evolutionary relationships among species within the Malvaceae family. The subtle differences in the pairwise heatmap analysis between the whole genome and the CDS region offer

significant insights into how information is extracted from various parts of the genome. The early branching and interspecific relationships within the genera *Gossypium* and *Hibiscus* clarify the evolutionary characteristics and timeline of these genera. In particular, the close relationship between *H. sinosyriacus* and *H. syriacus* suggests that these two species share a common recent ancestor and are evolutionarily proximate. Additionally, the classification of *T. amurensis* emphasizes how this species is evolutionarily unique compared with other species.

## 5. Conclusions

This study offers foundational insights into the structure and function of the *H. sinosyriacus* cp genome and establishes a basis for more in-depth research on its evolutionary position. We provide a comprehensive understanding of the cp genome structure of *H. sinosyriacus* and related species, whereas emphasizing the significance of gene positions within their respective boundaries. These structural variations and gene placements reflect the evolutionary traits and adaptations of each species. Such data are pivotal for phylogenetic and evolutionary studies of these taxa. Our findings shed light on the genetic relationships and evolutionary nuances of species within the *Hibiscus* genus. The numerous species-specific variations and characteristics identified through interspecific variation analysis will be useful for distinguishing species and developing various markers in the future. This study underscores the significance of the cp genome in understanding plant evolution and offers a foundation for future research in the Malvaceae family.

**Supplementary Materials:** The following supporting information can be downloaded at: https://www.mdpi.com/article/10.3390/f14112221/s1, Table S1: Variations of stop codons among *Hibiscus* spp.

**Author Contributions:** Conceptualization, methodology, software, formal analysis, visualization, and writing—original draft, S.-H.K.; validation, investigation, resources, and supervision, H.-Y.K.; supervision, writing—review and editing, and project administration, Y.-I.C.; supervision, review of the manuscript, and funding acquisition, H.S. All authors have read and agreed to the published version of the manuscript.

**Funding:** This research was funded by the National Institute of Forest Science, grant number FG0403-2023-02-2023.

**Data Availability Statement:** The data presented in this study have been deposited in the NCBI (https://ncbi.nlm.nih.gov/, accessed on 12 October 2023) GenBank with the accession number MZ_367751. The associated BioProject, BioSample, and SRA numbers are PRJNA789673, SAMN24146414, and SRR17253293, respectively. This data can be accessed at NCBI GenBank using the accession number MZ_367751.

**Conflicts of Interest:** The authors declare no conflicts of interest.

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
