# Peer review of "Comprehensive Analysis of Chloroplast Genome of Hibiscus sinosyriacus: Evolutionary Studies in Related Species and Genera"

_forests, doi:10.3390/f14112221_

Round 1

Reviewer 1 Report

Comments and Suggestions for Authors

Reviewer 2 Report

Comments and Suggestions for Authors

The manuscript under review is devoted to the analysis of the chloroplast genome of H. sinosyriacus and elucidate its evolutionary relationship with closely related species and genera. This work is of great interest, and I have several suggestions that authors may consider to improve the manuscript further.

1) On line 11, it says “the Hibiscus genus”, should be “The Hibiscus genus”.

2) On line 119, it says “nine species of Hibiscus, H. sinosyriacus, three species of Abelmoschus, and four species of Gossypium, with T. amurensis as the outgroup”, showing that the mumber of plants used for phylogenetic analysis is 18, but only 17 were found on line 122-128. Please confirm whether the quantity is correct. I think H. sinosyriacus should be one of the Hibiscus, which doesn't have to be listed separately.

3) On line 137, it says “he accession number MZ367751”, but on line 122-123, it says “H. sinosyriacus (MZ_367751)”, I think writing format should be uniform.

4) On line 146, it says “7 tRNAs”, but there are 8 names after this word. Please check whether the quantity is correct.

5) On line 310, the handwriting overlaps, please confirm whether it is correct.

6) On line 379, there is no introduction to Figure 7 in the paper, maybe it should be used in “3.7. Compararive phylogenetic analyses”.

7) In Table 4, “H. sinosyriacus” is right, while “H_sabdariffa” is wrong, The H is followed by a dot, not a horizontal line, the 8 names should be corrected.

8) On line 383, there are 17 species from 4 genera of the Malvaceas family. For plant names, italics are required, such as H. sinosyriacus, T. amurensis, and so on. It is better to modify the whole text.

9) On line 394, the picture of a and b from Figure 7, are too small, without detail on which plant that line corresponds to.

Comments on the Quality of English Language

Moderate editing of English language required

Reviewer 3 Report

Comments and Suggestions for Authors

Dear authors,

Thank you very much for adding new genomic data to the Hibiscus genus for a better understanding of the phylogenetic relationships of the Hibiscus genus of Malvaceae. The author provided a new chloroplast genome sequence of Hibiscus sinosyriacus for the first time. This data is useful for elucidating the phylogeny of H. sinosyriacus and related species in Malvaceae. This study is similar to previous studies of chloroplast genomes but it reported the complete chloroplast genome of H. sinosyriacus which is an essential resource for further genomic studies of Hibicus genus and Malvaceae family. Comparative analyses provide useful information for further studies on genetic populations and molecular markers. 

The conclusions are consistent with the evidence and arguments presented. The references are appropriate.

The content of the manuscript is suitable for Forests.

Below are my detailed comments:

1/ Could you please revise the title of the manuscript?

2/ Please use italic font for species names and gene names in the manuscript.

3/ There are more than 16 available chloroplast genome sequences of Malvaceae from NCBI. Therefore, please add the reason why the authors only used Hibiscus sinosyriacus and 16 other species for comparative analyses.

4/ Line 83 "Malvacae": please correct.

5/ Line 310: please check "Min" "Max" in the sentence.

-Figure 6: Please check "Min" and "Max" words because it is not in the figure but overlap the text.

Comments on the Quality of English Language

Some minor corrections should be made as mentioned in my comments. Thank you.

Reviewer 4 Report

Comments and Suggestions for Authors

Good work on sequencing and assembly of chloroplast genome, next in the series the authors are performing. Well done and presented.

I have the only minor suggestion for Figure 7: please add annotations to fig 7 A,B, to see if there are any rearrangements in the tree structure. You may use acronyms instead of full names.
